# SELF-ALIGNMENT OPTIMIZATION FOR LANGUAGE MODELS

## ABSTRACT

Traditional reinforcement learning from human feedback (RLHF) relies heavily on costly and time-consuming human-annotated datasets. Even Reinforcement Learning from AI Feedback (RLAIF), which trains a reward model using AI-generated preference data before refining the language model through reinforcement learning, remains expensive. These methods often necessitate either specialized reward model designs or larger models (e.g., GPT-4) for external labeling. In this paper, we introduce a dataset-free and annotation-free framework called Self-Alignment Optimization (SAO), which addresses the aforementioned issue by aligning the model with its own prompts and feedback as preferences. SAO begins with a chat-based model that engages in persona role-play to generate diverse prompts and responses, which are then self-evaluated and used for preference optimization. Extensive experiments with two strong LLMs on several benchmarks demonstrate the effectiveness of SAO. Specifically, on AlpacaEval 2.0, Gemma-2-9B-it-SAO achieves a Length-Controlled Win Rate (LC) of 69.2% and win rate (WR) of 66.0%, surpassing the baseline model by 18.1% and 27.9%. Llama-3-Instruct-8B-SAO reaches 33.3% LC and 39.0% WR, with performance improvements of 10.4% and 16.4%, respectively. On the MT-Bench benchmark, Gemma-2-9B-it-SAO and Llama-3-8B-Instruct-SAO score 7.41 and 6.76, compared to their pre-SAO scores of 7.09 and 6.70. The Arena-Hard benchmark shows even greater gains from SAO, with Gemma-2-9B-it's WR increasing from 52.6% to 70.1% and Llama-3-Instruct-8B's WR rising from 40.3% to 56.4%. In addition, our further experiments demonstrate that models fine-tuned with SAO exhibit similar or even superior performance on downstream NLP tasks compared to baseline models, rather than those trained with external labeled datasets, which enhance alignment ability but may compromise some general capabilities. We anticipate that this work will provide new insights for future research on self-improvement in LLMs. [1]

## 1 INTRODUCTION

Large language models (LLMs) have revolutionized the field of natural language processing (NLP), demonstrating remarkable capabilities in tasks such as mathematical reasoning, code generation, and dialogue generation (Cobbe et al., 2021; Wei et al., 2022; Bubeck et al., 2023; Chen et al., 2024b). A key advancement in LLMs is their alignment with human preference to create more helpful and reliable assistants (Mishra et al., 2021; Victor et al., 2022; Chung et al., 2022; Thoppilan et al., 2022). Common approaches include supervised fine-tuning (SFT) (Ouyang et al., 2022; Tunstall et al., 2023), based on human-demonstration pairs, and reinforcement learning from human feedback (RLHF) (Christiano et al., 2017; Ziegler et al., 2019; Stiennon et al., 2020; Bai et al., 2022a), which leverages signals from human preferences.

However, collecting demonstrations and preference labels is a expensive, time-consuming process, involving substantial human annotating efforts. To address this challenge, reinforcement learning from AI feedback (RLAIF) has been gaining attention, where a reward model is trained using AI-labeled preference data (Lee et al., 2024). However, RLAIF remains costly, typically requiring strong, proprietary models (*e.g.*, GPT-4) and specialized reward model designs (Jiang et al., 2023; Wang

---

[1] Our code is included in the supplementary material and is available at: https://anonymous.4open.science/r/SAO-ICLR2025-Submission.

et al., 2024) to work effectively. Moreover, these approaches often involve additional data-filtering procedures to obtain the final clean dataset (Xu et al., 2024b).

In this paper, we propose a dataset-free and annotation-free framework called Self-Alignment Optimization (SAO). Drawing inspiration from the compress-and-decompress approach to world knowledge from a persona perspective (Tseng et al., 2024; Ge et al., 2024; Chan et al., 2024) and the success of self-improvement mechanisms (Samuel, 2000; Chen et al., 2024b), SAO begins with a chat-based model and enables the LLM to engage in persona role-play to generate diverse prompts (*i.e.*, user queries). The LLM then generates paired responses and performs self-judgment to rank the responses. Lastly, preference optimization is employed to further refine the model. Our approach also aligns with the broader concept of model bootstrapping (Kearns & Valiant, 1994; Schapire, 1990; Freund, 1995; Freund & Schapire, 1997) and self-training (Vapnik, 1999; Grandvalet & Bengio, 2004; Lee, 2013).

Empirically, SAO demonstrates substantial performance gains across multiple benchmarks. On AlpacaEval 2.0, evaluated using GPT-4-Turbo-1106, Gemma-2-9B-it-SAO achieves a superior Length-Controlled Win Rate (LC) of 69.2% and a Win Rate (WR) of 66.0%. This performance surpasses the baseline Gemma-2-9B-it by 18.1% in LC and 27.9% in WR. Similarly, LLaMA-3-Instruct-8B-SAO shows substantial improvements, reaching 33.3% LC and 39.0% WR, which correspond to increases of 10.4% and 16.4% over its baseline. In the MT-Bench results, Gemma-2-9B-it-SAO and LLaMA-3-8B-Instruct-SAO achieve average scores of 7.41 and 6.70, respectively, compared to their baseline scores of 7.09 and 6.76. Furthermore, on the Arena-Hard benchmark, the win rate of Gemma-2-9B-it-SAO increases significantly from 52.6% to 70.1%, while LLaMA-3-Instruct-8B-SAO improves from 40.3% to 56.4%.

Moreover, SAO-tuned models either maintain or slightly enhance performance on objective downstream tasks, as evaluated on the Open LLM Leaderboard. Gemma-2-9B-it-SAO achieves an average score of 74.41 across all benchmarks, marginally surpassing its baseline score of 74.28. Similarly, LLaMA-3-8B-Instruct-SAO scores 68.20, slightly exceeding its baseline of 68.19. Unlike models trained on external labeled datasets, which may improve alignment ability at the cost of general performance, these results illustrate SAO's effectiveness in enhancing a model's subject-specific capabilities while preserving its downstream performance.

## 2 RELATED WORK

### 2.1 SYNTHETIC DATA FOR LLMS

In the context of SFT of LLMs, human-crafted data has proven remarkably effective, significantly enhancing performance on tasks like code generation (Roziere et al., 2023; Yang et al., 2023) and mathematical reasoning (Yuan et al., 2023; Luo et al., 2023). While human-generated data is typically of high quality, acquiring sufficient amounts is often prohibitively expensive. Consequently, the use of synthetic data has gained popularity as a cost-effective proxy for human data. This approach primarily leverages advanced LLMs, such as the GPT series (Radford et al., 2019; Brown et al., 2020; OpenAI, 2023), to generate high-quality data (Josifoski et al., 2023; Taori et al., 2023; Chiang et al., 2023; Li et al., 2023c). Recent studies have also emphasized the benefits of using LLMs' rephrasing capabilities to improve prompt responses (Deng et al., 2023; Prasad et al., 2023), as well as augmenting synthetic data for more effective SFT (Yu et al., 2023; Liu et al., 2023). Unlike prior research, which typically relies on more advanced models for generating synthetic data during pre-training or fine-tuning, our approach directly generates synthetic data from the target model itself, streamlining the process and reducing dependency on external resources.

### 2.2 LLM-AS-A-JUDGE

Using LLM-as-a-Judge prompting to evaluate language models has become a standard approach (Dubois et al., 2023; Li et al., 2023b; Fernandes et al., 2023; Bai et al., 2023; Saha et al., 2023). This technique is not only employed for evaluation but also for training reward models and curating data, as mentioned in prior works (Lee et al., 2023; Chen et al., 2024a; Li et al., 2024b). While some studies, such as Kim et al. (2023); Yuan et al. (2024b), focus on creating training data to enhance an LLM's performance as a judge, our approach uniquely integrates this judging capability with general instruction-following skills, setting it apart from existing methods.

## 2.3 Self-Play Language Models

Self-play (Samuel, 1959; Tesauro et al., 1995), where an algorithm learns by competing against itself, has gained significant attention for its effectiveness in multi-agent reinforcement learning (MARL). This method allows agents to interact with copies of themselves, progressively increasing the challenge and complexity of the learning environment. A seminal work in this area is AlphaGo Zero (Silver et al., 2017), which showcased remarkable performance against human players through a self-play learning scheme. Building on the success of self-play, subsequent research has explored various adaptations and implementations (Anthony et al., 2017; Lanctot et al., 2017; Bansal et al., 2018; Hernandez-Leal et al., 2018; Muller et al., 2019; Vinyals et al., 2019). Our method adopts a self-play optimization approach similar to AlphaGo Zero, where the model acts as its own judge to evaluate the responses it generates. This self-assessment enables the model to iteratively refine its outputs, improving its alignment and performance.

# 3 Problem Setting and Preliminaries

## 3.1 Generating Responses from a Chatbot

We consider an LLM parameterized by $\boldsymbol{\theta}$ and denoted as $p_{\boldsymbol{\theta}}$. The model takes an input sequence, $\mathbf{x} = [x_1, \ldots, x_n]$, referred to as the prompt, and generates a corresponding output sequence, $\mathbf{y} = [y_1, \ldots, y_m]$, as a response. The response $\mathbf{y}$ is sampled from the conditional probability distribution $p_{\boldsymbol{\theta}}(\cdot|\mathbf{x})$. In LLMs, $x_i$ and $y_j$ represent individual tokens from a predefined vocabulary, corresponding to the input and output sequences $\mathbf{x}$ and $\mathbf{y}$, respectively. The auto-regressive nature of the model $p_{\boldsymbol{\theta}}$ implies that it generates tokens sequentially, relying only on the sequence of previously generated tokens. Consequently, the model's generation process can be considered a Markov process. The conditional probability distribution $p_{\boldsymbol{\theta}}(\mathbf{y}|\mathbf{x})$ can be factorized as follows:

$$p_{\boldsymbol{\theta}}(\mathbf{y}|\mathbf{x}) = \prod_{j=1}^{m} p_{\boldsymbol{\theta}}(y_j|\mathbf{x}, \mathbf{y}_{<j}),$$

where $\mathbf{y}_{<1}$ is the empty sequence, and $\mathbf{y}_{<j} = [y_1, \ldots, y_{j-1}]$ for $j = 2, \ldots, m$.

## 3.2 RL Fine-tuning

RL fine-tuning (Christiano et al., 2017; Bai et al., 2022a; Gao et al., 2023) is an alternative method for enhancing the specific capabilities of general-purpose pre-trained models. Typically, RL fine-tuning is employed after SFT to achieve better alignment in LLMs (Tunstall et al., 2023). For a given sequence pair $(\mathbf{x}, \mathbf{y})$, RL fine-tuning requires a deterministic reward function $r(\cdot, \cdot)$, where a higher reward $r(\mathbf{x}, \mathbf{y})$ indicates a better response $\mathbf{y}$ to the given prompt $\mathbf{x}$. The objective of RL fine-tuning is to maximize the following function:

$$L_{\mathrm{RL}}(\boldsymbol{\theta}) = \mathbb{E}_{\mathbf{x} \sim q(\cdot), \mathbf{y} \sim p_{\boldsymbol{\theta}}(\cdot|\mathbf{x})}[r(\mathbf{x}, \mathbf{y})] - \lambda \mathbb{E}_{\mathbf{x} \sim q(\cdot)} \mathrm{KL}\big(p_{\boldsymbol{\theta}}(\cdot|\mathbf{x}) \big| p_{\mathrm{ref}}(\cdot|\mathbf{x})\big),$$

where the Kullback-Leibler (KL) regularization term ensures that the policy model $p_{\boldsymbol{\theta}}$ remains close to the reference model $p_{\mathrm{ref}}$. The regularization parameter $\lambda > 0$ controls the extent to which the policy model can deviate from the reference model. In practice, the reference model $p_{\mathrm{ref}}$ is often initialized from the SFTed model. KL regularization is crucial for preventing excessive deviation from the reference model, thereby reducing the risk of mode collapse.

A significant challenge in RL fine-tuning is designing a robust reward function. This function typically requires training on a preference dataset, which demands substantial resources. This process often involves comprehensive evaluations by either human annotators, known as reinforcement learning from human feedback (RLHF) (Christiano et al., 2017; Bai et al., 2022a), or by strong AI agents, referred to as reinforcement learning from AI feedback (RLAIF) (Bai et al., 2022b).

# 4 Self-Alignment Optimization

We present the overview framework of Self-Alignment Optimization (SAO) in Algorithm 1. This approach involves creating diverse prompts based on different personas and generating pairwise

responses, which are subsequently ranked according to their quality. The ranked pairs form a dataset for preference optimization, facilitating model improvement without the need for human-labeled data. We will discuss this process in detail in the following sections.

---

**Algorithm 1** Self-Alignment Optimization (SAO)

---

**Require:** Base model $\mathcal{M}_{\theta_0}$, number of personas $n$, preference optimization algorithm $\mathcal{A}$
**Ensure:** Optimized model $\mathcal{M}_{\theta_1}$
 1: Initialize personas templates $\{r_1, r_2, \ldots, r_n\}$
 2: Initialize dataset $\mathcal{D} \leftarrow \emptyset$
 3: **for** $i = 1, 2, \ldots, n$ **do**
 4:     Generate prompt: $x_{\text{prompt}}^i \leftarrow \mathcal{M}_{\theta_0}(r_i)$                       $\triangleright$ § 4.1
 5:     Generate responses: $y_1, y_2 \leftarrow \mathcal{M}_{\theta_0}(\cdot \mid x_{\text{prompt}}^i)$           $\triangleright$ § 4.2
 6:     Rank responses: $(y_{\text{w}}, y_{\text{l}}) \leftarrow \mathcal{R}_{\theta_0}(y_1, y_2 \mid x_{\text{prompt}}^i, x_{\text{rank}})$    $\triangleright$ § 4.3
 7:     $\mathcal{D} \leftarrow \mathcal{D} \cup \{(x_{\text{prompt}}^i, y_{\text{w}}, y_{\text{l}})\}$                     $\triangleright$ § 4.4
 8: **end for**
 9: Optimize: $\theta_1 \leftarrow \underset{\theta}{\arg\min} \, \mathcal{L}_{\mathcal{A}}(\mathcal{M}_\theta)$                   $\triangleright$ § 4.5
10: **return** $\mathcal{M}_{\theta_1}$

---

## 4.1 DIVERSE PROMPT GENERATION

To facilitate a comprehensive range of training scenarios, we utilize a LLM denoted as $\mathcal{M}$, parameterized by $\theta$, for the generation of diverse prompts tailored to specific persona roles, as depicted in Figure 1 (top). Given a set of roles templates $\mathcal{R} = \{r_i\}_{i=1}^n$, we derive a unique prompt for each persona:

$$x_{\text{prompt}}^i = \mathcal{M}_\theta(r_i) \tag{4.1}$$

In this context, $x_{\text{prompt}}^i$ represents the prompt generated for the $i$-th persona $r_i$. To ensure the diversity of generated prompts, we impose a constraint such that each persona can generate only a single question. The persona resources are randomly sampled from Persona-Hub, which encompasses approximately 200,000 entries, as constructed by Chan et al. (2024).

## 4.2 PAIR-WISE RESPONSE GENERATION

For each generated prompt, we create a pair of responses to enable comparative evaluation. Let $\mathcal{X}$ be the space of prompts and $\mathcal{Y}$ the space of responses. For each prompt $x_{\text{prompt}}^i \in \mathcal{X}$, we generate two responses $y_1, y_2 \in \mathcal{Y}$ using the $\mathcal{M}_\theta$:

$$y_1, y_2 \sim \mathcal{M}_\theta(\cdot \mid x_{\text{prompt}}^i) \tag{4.2}$$

Generating additional responses could potentially yield better performance but would increase computational costs and evaluation time. We leave this exploration for future work.

## 4.3 SELF-JUDGMENT

To assess the quality of generated responses, we implement a self-judgment mechanism. This process entails the LLM evaluating its own outputs, thereby simulating human preferences. As illustrated in Figure 1 (bottom), we query the LLM with a ranking prompt $x_{\text{rank}}$ to compare the responses $y_1$ and $y_2$ based on their relevance and quality relative to $x_{\text{prompt}}^i$:

$$(y_{\text{w}}, y_{\text{l}}) = \mathcal{R}_\theta(y_1, y_2 \mid x_{\text{prompt}}^i, x_{\text{rank}}) \tag{4.3}$$

Here, $y_{\text{w}}$ and $y_{\text{l}}$ represent the superior and inferior responses, respectively. The function $\mathcal{R}_\theta$ encapsulates the LLM's decision-making process in ranking the responses.

---

**Persona Instruction Example**

Guess a prompt that the following persona may ask you to do:

A Political Analyst specialized in El Salvador's political landscape.

**Note:**
1. The prompt should be informative and specific.
2. Your output should start with "User prompt:"

---

**Pair-wise Response Ranking**

You are an impartial judge. Your task is to rank two answers to a given prompt based on their quality.

**Prompt:** {prompt}

**Response 1:** `<Response 1>` {response 1} `</Response 1>`

**Response 2:** `<Response 2>` {response 2} `</Response 2>`

Please carefully read each response and evaluate them based on the following criteria:
1. Relevance and specificity to the prompt
2. Accuracy and correctness of information
3. Completeness and comprehensiveness
4. Clarity and understandability

Then, rank these two responses from best to worst. You must output your ranking strictly in the following format: **ranking: X > Y**, where X and Y represent one of 1 or 2, without repetition.

Remember, you must output a complete ranking including both options. Now, please provide your ranking:

---

Figure 1: The top box displays the persona instruction prompt, which directs the LLM to generate a specific prompt based on a given persona. The bottom box illustrates the pair-wise response ranking prompt, instructing the LLM to compare and rank responses based on specific criteria modified from Shen et al. (2024).

## 4.4 DATASET CONSTRUCTION

We construct a synthetic dataset $\mathcal{D}$ by aggregating the generated prompts and ranked responses for each persona:

$$\mathcal{D} = \{(x^i_{\text{prompt}}, y^i_{\text{win}}, y^i_{\text{loss}})\}^n_{i=1} \tag{4.4}$$

where $n$ is the total number of personas. This dataset forms the cornerstone of our preference optimization process, allowing the model to learn from its own generated and ranked responses across diverse personas.

## 4.5 PREFERENCE OPTIMIZATION

Recent advancements in preference optimization have demonstrated significant potential in aligning LLMs with human preferences. Techniques such as Direct Preference Optimization (DPO) (Rafailov et al., 2023) and Simple Preference Optimization (SimPO) (Meng et al., 2024) have gained prominence due to their efficacy in fine-tuning LLMs to better reflect human preferences. In this study, we employ SimPO due to its suitability for our dataset, which frequently contains longer responses. SimPO's length normalization technique effectively captures nuanced information at the token level, making it

particularly well-suited to our requirements and we provide a more detailed analysis and comparison of these methods in Section 5.4.2.

SimPO introduces a length-normalized reward formulation that aligns with the likelihood metric guiding generation with a scaling constant $\beta$:

$$r(x, y) = \frac{\beta}{|y|} \log \mathcal{M}_\theta(y \mid x) = \frac{\beta}{|y|} \sum_{i=1}^{|y|} \log \mathcal{M}_\theta(y_i \mid x, y_{<i}) \tag{4.5}$$

Additionally, it incorporates a target reward margin $\gamma > 0$ to ensure a minimum difference between the rewards of winning and losing responses:

$$p(y_w \succ y_l \mid x) = \sigma\left(r(x, y_w) - r(x, y_l) - \gamma\right) \tag{4.6}$$

The overall objective is then formulated as:

$$\mathcal{L}(\mathcal{M}_\theta) = -\mathbb{E}_{(x, y_w, y_l) \sim \mathcal{D}} \left[ \log \sigma \left( \frac{\beta}{|y_w|} \log \mathcal{M}_\theta(y_w|x) - \frac{\beta}{|y_l|} \log \mathcal{M}_\theta(y_l|x) - \gamma \right) \right] \tag{4.7}$$

This objective function guides the optimization process, enabling the model to learn from its self-generated preferences and improve its alignment with desired outcomes.

## 5 EXPERIMENTS

### 5.1 EXPERIMENT SETTING

In our experiments, we use the Gemma-9B-it model as the base and apply a similar fine-tuning process to Llama-3-8B-Instruct. To ensure diverse prompts and responses, we set the temperature to 0.6 and utilize VLLM for accelerated generation. For preference optimization, we incorporate Flash Attention 2 and bfloat16 precision, with hyperparameters set to $\beta = 10$ and $\gamma = 3$. Additionally, we employ DeepSpeed with ZeRO-3 optimization for effective memory management and scalability. All experiments are conducted over a single epoch with a global batch size of 128 across four A100 GPUs. The learning rate is set to $1 \times 10^{-6}$, following a cosine decay scheduler with a warmup ratio of 0.1. We use a synthetic dataset of 60k samples as the default setting for self-alignment optimization.

### 5.2 EVALUATION METRICS AND BASELINES

**Evaluation Metrics.** Our experimental evaluation employs a comprehensive set of metrics to assess model performance across various dimensions. For subjective benchmarks, we primarily focus on AlpacaEval 2 (Li et al., 2023a), an LLM-based automatic evaluation benchmark utilizing prompts from AlpacaFarm (Dubois et al., 2024). In this benchmark, model responses and GPT-4-Turbo generated reference responses are evaluated by GPT-4-Turbo or Qwen2-72B-Instruct annotators. We also incorporate GPT-4o-mini to evaluate two additional subjective benchmarks: Arena-Hard (Li et al., 2024a), an automatic evaluation tool featuring 500 challenging user queries, and MT-Bench (Zheng et al., 2023), a set of 80 high-quality multi-turn open-ended questions covering topics such as writing, role-playing, math, and coding. For objective benchmarks, we utilize the Open LLM Leaderboard (Beeching et al., 2023), which comprises six datasets focusing on various aspects of language model evaluation, including math problem-solving, language understanding, human falsehood mimicking, and reasoning. We adhere to the standard evaluation process, using in-context learning to prompt the models and compute the average score across these six datasets to measure performance comprehensively.

**Baselines.** In our comparisons, we include a diverse set of baselines. These encompass vanilla models such as GPT-4o-05-13, Claude-3.5-Sonnet, and GPT4-Turbo-04-09. Additionally, we evaluate models trained on external labeled datasets, like Llama-3-Instruct-8B-SimPO (Meng et al., 2024), which has been fine-tuned using the Ultrafeedback dataset (Cui et al., 2024) for preference optimization.We also consider Self-Rewarding-70B-Iter3 (Yuan et al., 2024a), which is trained using a mixture of external labeled datasets and synthetic data. Additionally, we examine Gemma-2-9B-SPPO-Iter3 (Wu et al., 2024), which generates responses based on Ultrafeedback prompts and utilizes preference pairs

Table 1: Comparative analysis of various baseline models and our proposed SAO method using AlpacaEval 2.0. The table presents Length-Controlled Win Rate (LC), Win Rate (WR), and Standard Deviation (STD) for each model, evaluated against GPT-4-Turbo-1106 and Qwen2-72B-Instruct.

| Model | AlpacaEval 2.0 | | | | | |
| | GPT-4-Turbo-1106 | | | Qwen2-72B-Instruct | | |
| | LC (%) | WR (%) | STD | LC (%) | WR (%) | STD |
|---|---|---|---|---|---|---|
| *Vanilla Models* | | | | | | |
| Llama-3-8B-Instruct | 22.9 | 22.6 | 1.3 | 29.4 | 29.2 | 1.6 |
| Yi-34B-Chat | 27.2 | 29.7 | 1.3 | 33.3 | 37.0 | 1.7 |
| GPT-4-Turbo-04-09 | 55.0 | 46.1 | 1.5 | 49.0 | 39.1 | 1.7 |
| Gemma-2-9B-it | 51.1 | 38.1 | - | 56.5 | 39.3 | 1.7 |
| Claude-3.5-Sonnet | 52.4 | 40.6 | 1.5 | **56.8** | 40.5 | 1.7 |
| GPT-4o-05-13 | **57.5** | **51.3** | 1.5 | 51.8 | **44.7** | 1.8 |
| *Models Trained Using External Labeled Dataset* | | | | | | |
| Self-Rewarding-70B-Iter3 (Yuan et al., 2024a) | - | 20.4 | - | - | - | - |
| Llama-3-Instruct-8B-SimPO (Meng et al., 2024) | 53.7 | 47.5 | - | 54.2 | 45.9 | 1.8 |
| Gemma-2-9B-SPPO-Iter3 (Wu et al., 2024) | 53.3 | 47.8 | - | - | - | - |
| Gemma-2-9B-it-SimPO (Meng et al., 2024) | **72.4** | **65.9** | 1.4 | **74.5** | **65.5** | 1.7 |
| *Models Trained Only Using Self-Synthetic Dataset* | | | | | | |
| Llama-3-8B-Magpie-SFT-v0.1 (Xu et al., 2024b) | 24.2 | 25.2 | - | 26.2 | 29.2 | 1.6 |
| Llama-3-Instruct-8B-SAO (Ours) | 33.3 (+10.4) | 39.0 (+16.4) | 1.4 | 42.3 (+12.9) | 49.1 (+19.9) | 1.8 |
| Gemma-2-9B-it-SAO (Ours) | **69.2** (+18.1) | **66.0** (+27.9) | 1.4 | **76.0** (+19.5) | **71.6** (+32.3) | 1.6 |

labeled by external tools. Furthermore, we compare against the recently developed self-synthetic baseline, Llama-3-8B-Magpie-SFT-v0.1 (Xu et al., 2024b), which was originally trained on synthetic SFT pair data generated by the model itself to improve its alignment capability. This diverse set of baselines allows for a comprehensive evaluation of our SAO method against both traditional and innovative fine-tuning approaches.

## 5.3 RESULTS

### 5.3.1 PERFORMANCE ON ALPACAEVAL 2.0

Our SAO-tuned models demonstrate substantial performance improvements on AlpacaEval 2.0 when evaluated by both GPT-4-Turbo-1106 and Qwen2-72B-Instruct. When assessed by GPT-4-Turbo-1106, Gemma-2-9B-it-SAO achieves a Length-Controlled Win Rate (LC) of 69.2% and a Win Rate (WR) of 66.0%, representing increases of 18.1% and 27.9% respectively over the baseline Gemma-2-9B-it (51.1% LC, 38.1% WR). This performance surpasses all vanilla models, including the top-performing GPT-4o (05-13) at 57.5% LC and 51.3% WR. Moreover, Gemma-2-9B-it-SAO competes closely with models trained on external datasets, approaching the performance of Gemma-2-9B-it-SimPO (72.4% LC, 65.9% WR). Similarly, Llama-3-Instruct-8B-SAO exhibits significant improvements, reaching 33.3% LC and 39.0% WR, increases of 10.4% and 16.4% respectively over the baseline Llama-3-8B-Instruct (22.9% LC, 22.6% WR). When evaluated by Qwen2-72B-Instruct, Gemma-2-9B-it-SAO continues to excel, achieving 76.0% LC and 71.6% WR. These represent substantial improvements of 19.5% and 32.3% over the baseline Gemma-2-9B-it (56.5% LC, 39.3% WR) and even outperform models trained on external datasets, such as Gemma-2-9B-it-SimPO (74.5% LC, 65.5% WR). Llama-3-Instruct-8B-SAO also demonstrates significant improvement when evaluated by Qwen2-72B-Instruct, reaching 42.3% LC and 49.1% WR, increases of 12.9% and 19.9% over Llama-3-8B-Instruct (29.4% LC, 29.2% WR). These results underscore the efficacy of our SAO method in enhancing model performance across different base models and evaluation metrics, for both LC and WR. Notably, our approach achieves these improvements without relying on external labeled datasets, highlighting its potential for efficient and scalable model enhancement.

### 5.3.2 MT-BENCH AND ARENA-HARD PERFORMANCE

Our evaluation extended to two other mainstream subjective benchmarks, MT-Bench and Arena-Hard, yielding compelling results that underscore the efficacy of SAO fine-tuning. As shown in Figure 2 (left), on the MT-Bench benchmark, Gemma-2-9B-it-SAO achieved an average score of 7.41, surpassing the baseline Gemma-2-9B-it (7.09) by 0.32 points. Similarly, LLaMA-3-8B-Instruct-SAO

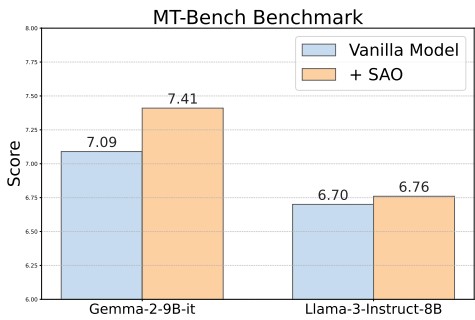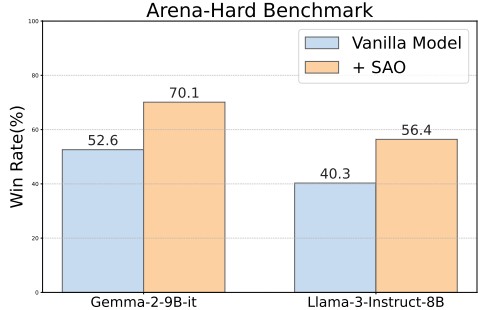

Figure 2: Performance comparison on MT-Bench and Arena-Hard following SAO fine-tuning.

Table 2: Performance comparison of models on downstream NLP benchmarks from the Open LLM Leaderboard. The values in parentheses indicate the number of few-shot examples (shots).

| Model | ARC (25) | TruthfulQA (0) | Winograd (5) | GSM8K (5) | HellaSwag (10) | MMLU (5) | Average |
|---|---|---|---|---|---|---|---|
| Gemma-2-9B-it-SAO | 71.50 | 62.76 | 77.35 | 80.29 | 82.53 | 72.02 | **74.41** |
| Gemma-2-9B-it-SimPO | 69.11 | 59.00 | 73.72 | 81.96 | 66.65 | 71.82 | 70.38 |
| Gemma-2-9B-it | 71.08 | 60.15 | 78.06 | 82.34 | 81.73 | 72.30 | 74.28 |
| Llama-3-8B-Instruct-SAO | 63.57 | 49.58 | 74.66 | 76.72 | 78.96 | 65.72 | **68.20** |
| Llama-3-8B-Instruct-SimPO | 66.64 | 63.86 | 74.74 | 55.65 | 78.97 | 66.51 | 67.73 |
| Llama-3-8B-Instruct | 61.95 | 51.70 | 75.30 | 75.66 | 78.78 | 65.72 | 68.19 |

reached an average of 6.76, improving upon the baseline LLaMA-3-8B-Instruct (6.70) by 0.06 points, demonstrating an enhanced ability to handle multi-turn open-ended questions. The Arena-Hard benchmark revealed even more substantial performance gains, as shown in Figure 2 (right), with Gemma-2-9B-it experiencing a remarkable increase in WR from 52.6% to 70.1% after SAO tuning, marking a 17.5 percentage point improvement. Meanwhile, LLaMA-3-Instruct-8B's WR rose from 40.3% to 56.4%, reflecting a 16.1 percentage point increase. These significant advancements in the Arena-Hard benchmark highlight the effectiveness of SAO tuning in enhancing model performance on diverse and challenging tasks.

### 5.3.3 DOWNSTREAM EVALUATION ON OPEN LLM LEADERBOARD

To assess the impact of our proposed SAO method on downstream task performance, we conducted a comprehensive evaluation across diverse tasks using the Open LLM Leaderboard benchmarks, as detailed in Table 2. The results demonstrate that SAO-tuned models generally maintain or slightly improve their capabilities compared to their baseline counterparts.

For the Gemma-2-9B series, our SAO-tuned version achieves an average score of 74.41 across all benchmarks, marginally surpassing the baseline Gemma-2-9B-it (74.28). Notably, Gemma-2-9B-it-SAO shows improvements in ARC (+0.42), TruthfulQA (+2.61), and HellaSwag (+0.80) tasks, while maintaining comparable performance in others. Similarly, Llama-3-8B-Instruct-SAO (68.20) slightly outperforms its baseline (68.19), with notable enhancements in ARC (+1.62) and HellaSwag (+0.18) tasks. Interestingly, models optimized with external datasets, such as Gemma-2-9B-it-SimPO and Llama-3-8B-Instruct-SimPO, while achieving impressive results on alignment tasks, show a decrease in overall performance across these general benchmarks. Gemma-2-9B-it-SimPO's average score (70.38) is significantly lower than both the baseline and SAO-tuned versions, with notable declines in Winograd (-4.34) and HellaSwag (-15.08) tasks. Llama-3-8B-Instruct-SimPO, despite improvements in certain areas like TruthfulQA (+12.16), also shows a slight overall decrease (67.73) compared to its baseline, primarily due to a substantial drop in GSM8K performance (-20.01).

We hypothesize that this performance discrepancy stems from the nature of externally annotated datasets, which may not align perfectly with the current capabilities of these language models. While such datasets can yield improvements in specific alignment tasks, they may inadvertently compromise the model's general abilities. In contrast, our SAO method, utilizing self-generated data, appears to more accurately represent and enhance the model's intrinsic capabilities, leading to consistent performance across a wide range of tasks without significant trade-offs.

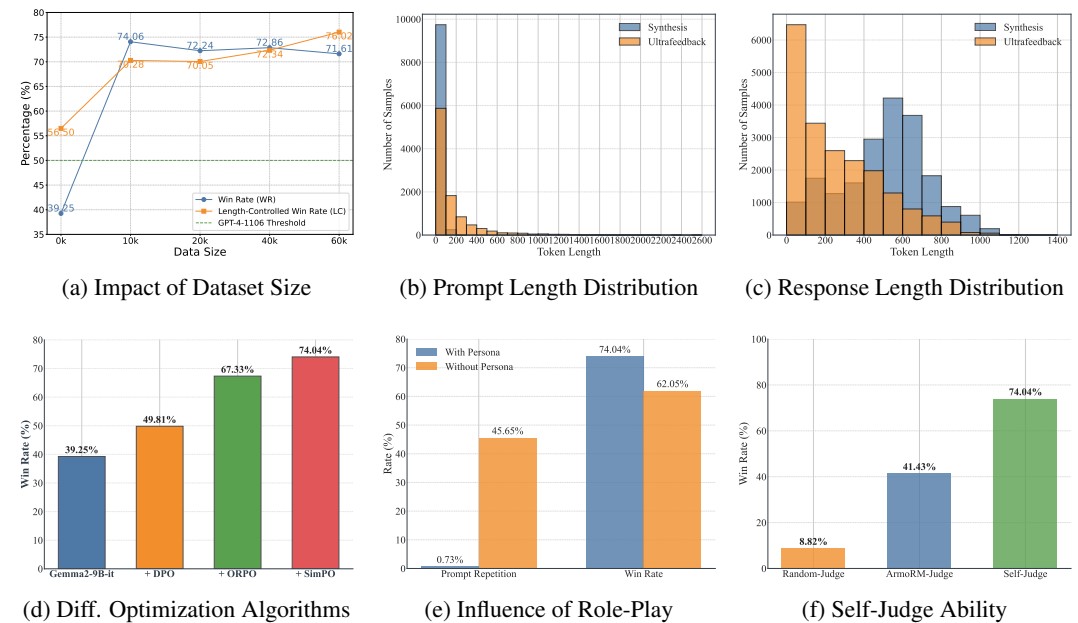

(a) Impact of Dataset Size  (b) Prompt Length Distribution  (c) Response Length Distribution

(d) Diff. Optimization Algorithms  (e) Influence of Role-Play  (f) Self-Judge Ability

Figure 3: Further exploration of the self-alignment optimization from various perspectives.

## 5.4 FURTHER EXPLORATION

As shown in Figure 3, we utilized Gemma-2-9B-it-SAO to further explore various factors influencing model performance on Alpacaeval 2.0, evaluated by Qwen2-72B-Instruct. This evaluation includes the impact of dataset size. For other factors, such as optimization algorithms, persona role-play, and judging methods, we observed that even with the 10k dataset, the SAO-tuned model achieved promising improvements. Consequently, we focused on these aspects using the 10k synthetic dataset to make the evaluation process more cost-effective.

### 5.4.1 IMPACT OF SYNTHETIC DATASET SIZE

As illustrated in Figure 3a, the performance of Gemma-2-9B-it-SAO improved significantly with an increase in synthetic dataset size. The WR rose from 39.25% for the vanilla model (0k) to 74.06% with a 10k dataset, stabilizing around 72% for larger datasets. Additionally, the LC improved, reaching 76.02% with a 60k dataset. Interestingly, we found that even a small amount of self-alignment data can significantly enhance model alignment performance.

### 5.4.2 DIFFERENT OPTIMIZATION ALGORITHMS

To investigate the influence of different optimization algorithms, we compared three mainstream approaches: DPO (Xu et al., 2024a), ORPO (Hong et al., 2024), and SimPO (default) (Meng et al., 2024). Figure 3d illustrates the performance of these algorithms. Starting from the baseline Gemma-2-9B-it model (39.25% WR), we observed progressive improvements: DPO raised the WR to 49.81%, ORPO increased it further to 67.33%, and SimPO achieved the highest WR of 74.04%. The superior performance of SimPO may be attributed to the characteristics of our generated dataset, as shown in Figures 3b and 3c. Compared to external labeled datasets, our synthetic dataset tends to generate shorter prompts and longer responses, making SimPO's length normalization particularly effective in this context. Examples of the synthetic dataset are provided in Table 3 in the Appendix.

### 5.4.3 INFLUENCE OF PERSONA ROLE-PLAY

A key component of our method is the persona role-play, which enhances the diversity of prompt generation. Figure 3e illustrates its impact. With this mechanism, the model achieved a WR of 74.04% and a significantly lower prompt repetition rate of 0.73%. In contrast, without persona role-play, the

WR decreased to 62.05%, and the prompt repetition rate rose to 45.65%. These results underscore the critical role of persona role-play in enhancing model performance and reducing repetition. Notably, even with a high number of repetitive prompts, the overall framework demonstrated significant improvements compared to the vanilla model, which achieved only a 39.3% WR. We attribute this robustness to our SAO algorithm, which enables the model to self-improve even in the presence of a redundant dataset. And some repetitive examples are listed in Table 4 in the Appendix.

### 5.4.4 SELF-JUDGE ABILITY

To evaluate the model's ability to assess its own responses, we examined three settings: (1) Random-Judge, where responses were randomly selected from generated pairs; (2) ArmoRM-Judge, using the state-of-the-art external labeling tool ArmoRM-Llama3-8B-v0.1 (Wang et al., 2024) to rank responses; and (3) Self-Judge, where the vanilla model evaluated prompts using our proposed pairwise-ranking method. As shown in Figure 3f, the Self-Judge mechanism outperformed the others with a win rate of 74.04%, significantly surpassing ArmoRM-Judge (41.43%) and Random-Judge (8.82%). The poor performance of Random-Judge underscores the necessity of a meaningful evaluation process. The strong results for Self-Judge suggest that the model has a robust ability to assess its own responses. Although ArmoRM outperformed Random-Judge, it still fell short of Self-Judge, likely because its broader training data distributions were not fully aligned with the model's specific capabilities.

## 6 CONCLUSION

In this paper, we introduce a dataset-free and annotation-free framework, Self-Alignment Optimization (SAO), for fine-tuning large language models (LLMs) using only synthetic data pairs generated by the models themselves. This approach eliminates the need for human-labeled datasets or external labeling tools, relying instead on external signals from existing personas. Remarkably, SAO achieves significant performance improvements across various benchmarks, including AlpacaEval 2.0, Arena-Hard, and MT-Bench, consistently outperforming baseline models. Furthermore, it demonstrates robustness by maintaining or even enhancing performance on downstream NLP tasks. We believe this straightforward and effective post-training strategy has the potential to unlock the latent capabilities of LLMs and provide valuable insights for future research on self-improvement in these models.

## 7 LIMITATIONS

While our experimental results are promising, this study is constrained by the use of models smaller than 10 billion parameters due to resource limitations. We anticipate that scaling the SAO framework to larger models could yield even greater performance enhancements. Additionally, although our approach has demonstrated effectiveness with simple prompt templates, investigating more complex templates may provide further improvements. Future research should address these areas to fully leverage the potential of the SAO framework.

## 8 SOCIAL IMPACT

The introduction of the SAO framework offers a valuable solution to the challenges of fine-tuning LLMs without extensive external supervision. This approach can significantly reduce the manual effort and time required for model training, thereby enhancing accessibility to NLP technologies for individuals and organizations with limited resources. However, it is crucial to exercise caution with this self-improvement framework, as it relies entirely on self-synthesized datasets, which may lead to the generation of inaccuracies or hallucinations in certain cases.

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

# A   APPENDIX

| | |
|---|---|
| **Input Persona** | A newly hired general counsel at TurpCo Industries |
| **Generated Prompt** | As a newly hired General Counsel at TurpCo Industries, I need to understand the legal landscape of our operations. Could you provide a detailed report on the major legal risks and compliance issues our company might face in the next fiscal year, including any potential changes in legislation that could affect our business? Additionally, please include recommendations for mitigating these risks and ensuring our compliance with all relevant laws and regulations. |
| **Response 1** | **Executive Summary:**
As the General Counsel of TurpCo Industries, it is essential to understand the legal risks and compliance issues that our company may face in the next fiscal year. This report provides an overview of the major legal risks and compliance issues that our company may face, including potential changes in legislation that could affect our business. The report also includes recommendations for mitigating these risks and ensuring our compliance with all relevant laws and regulations.
**Legal Risks and Compliance Issues:**
1. **Data Privacy and Cybersecurity:** As the world becomes increasingly digital, data privacy and cybersecurity risks have become more prevalent. Our company must ensure that we comply with all relevant data protection laws and regulations, such as the General Data Protection Regulation (GDPR) in Europe and the California Consumer Privacy Act (CCPA) in the United States. We must also have robust cybersecurity measures in place to protect our company's and our customers' data...... |
| **Response 2** | **Subject: Legal Risk Assessment and Compliance Report for TurpCo Industries**
Dear [Recipient's Name],
I am pleased to present a comprehensive report on the major legal risks and compliance issues that TurpCo Industries may face in the upcoming fiscal year. This report will also address potential legislative changes that could impact our operations, along with recommendations for risk mitigation and ensuring legal compliance.
1. **Overview of Legal Risks and Compliance Issues:**
a. **Environmental Regulations:**
- **Risk:** TurpCo Industries' manufacturing processes may be subject to stringent environmental regulations, including those related to emissions, waste disposal, and water usage.
- **Potential Changes:** The Environmental Protection Agency (EPA) is considering updating regulations to reduce greenhouse gas emissions and tighten waste disposal standards..... |
| **Ranking** | 2 > 1 |

Table 3: Example of Synthetic Dataset

| Prompt | Repetition Count |
|---|---|
| Summarize the main plot points of the novel "Pride and Prejudice" by Jane Austen, focusing on the development of the relationship between Elizabeth Bennet and Mr. Darcy. | 267 |
| Summarize the main plot points of the novel "Pride and Prejudice" by Jane Austen, focusing on the relationship between Elizabeth Bennet and Mr. Darcy. | 258 |
| Write a short story (around 500 words) about a young inventor who creates a device that can translate animal languages, but struggles with the ethical implications of their invention. | 178 |
| Summarize the main plot points of the novel *Pride and Prejudice* by Jane Austen, focusing on the relationship between Elizabeth Bennet and Mr. Darcy. | 114 |
| Summarize the main plot points of the novel *Pride and Prejudice* by Jane Austen, focusing on the development of the relationship between Elizabeth Bennet and Mr. Darcy. | 105 |

Table 4: Top Repeated Prompts Without Role Play in 10k Synthesized Dataset

| **Instruction** |
| What are the names of some famous actors that started their careers on Broadway? |
| **Gemma2-9B-it Output** |
| Here are some famous actors who started their careers on Broadway:
**Legends:**
* **Meryl Streep:** Made her Broadway debut in 1975 and won a Tony Award for "The Cherry Orchard."
* **Denzel Washington:** Won a Tony Award for his performance in "Fences" and has starred in many other Broadway productions.
* **Al Pacino:** Rose to fame on stage before transitioning to film, winning a Tony Award for "Does a Tiger Wear a Necktie?"
* **Robert De Niro:** Made his Broadway debut in 1964 and has since returned for several productions.
* **James Earl Jones:** Known for his iconic voice, he won a Tony Award for "Fences" and has been a mainstay on Broadway for decades.
**More Recent Stars:**
* **Lin-Manuel Miranda:** Created and starred in the smash hit "Hamilton," winning multiple Tony Awards for his work.
* **Andrew Garfield:** Earned Tony Award nominations for "Death of a Salesman" and "Angels in America."
* **Scarlett Johansson:** Made her Broadway debut in "A View from the Bridge" and has since returned for other productions.
* **Nathan Lane:** A Tony Award-winning actor known for his comedic roles in musicals like "The Producers" and "Guys and Dolls."
* **Hugh Jackman:** Known for his role as Wolverine, he has also starred in several Broadway musicals, winning a Tony Award for "The Boy From Oz."
This is just a small sample of the many talented actors who have honed their craft on Broadway. Let me know if you'd like to explore actors from a specific genre or era! |
| **Gemma2-9B-it-SAO Output** |
| Here's a list of famous actors, spanning generations and genres, who honed their craft and gained initial recognition on Broadway:
**Stage Legends & Classics:**
* **Marlon Brando, James Dean, Montgomery Clift:** Pioneered Method acting, rose to fame through iconic stage productions before Hollywood.
* **Humphrey Bogart, Katharine Hepburn, Laurence Olivier, Ingrid Bergman:** Established themselves as stage stars before transitioning to film.
* **Judy Garland, Ethel Merman, Carol Channing, Yul Brynner:** Musical theatre icons who built enduring careers on Broadway.
* **Angela Lansbury, Julie Andrews, Audra McDonald:** Multiple Tony Award winners with long, distinguished Broadway careers.
**Contemporary Stars:**
* **Denzel Washington, Viola Davis, Nathan Lane, Patrick Stewart:** Highly acclaimed actors with both stage and screen success.
* **Tom Hanks, Meryl Streep, Daniel Day-Lewis, Michelle Williams:** Oscar winners who began their careers in theatre.
* **Idina Menzel, Lin-Manuel Miranda, Andrew Garfield, Jake Gyllenhaal:** Known for both stage and screen, often returning to Broadway.
* **Hugh Jackman, Bradley Cooper, Emma Stone, Scarlett Johansson:** Made their mark on Broadway before achieving major film success.
**Notable Mentions:**
* **Robert De Niro, Al Pacino, Christopher Walken:** Strong stage backgrounds, though perhaps less known for their Broadway work specifically.
This list is by no means exhaustive, but it highlights the rich history of actors who found their footing and developed their skills on the Broadway stage. |

Table 5: Outputs from Gemma2-9B-it and Gemma2-9B-it-SAO models

