# OpenReview forum: "Self-Alignment Optimization for Language Models"
_ICLR.cc/2025/Conference — Submitted to ICLR 2025_

### Official Review · Reviewer_acKt · 2024-11-03

**Soundness:** 2
**Presentation:** 3
**Contribution:** 2
**Rating:** 3
**Confidence:** 4

**Summary:**

Paper proposes Self-Alignmenet Optimization(SAO) claiming it is a data-free, annotation-free framework to perform preference alignment.
The paper shows that without using an external prompt space, and an external reward model, LLM can learn from an on-policy sampled dataset annotated by using itself as a judge. The paper demonstrates improvements on 3 trusted alignment Benchmarks, AlpacaEval, MT-Bench, and ArenaHard over the starting model, and for gemma model in particular, it shows comparable performance as models aligned using SoTA trained reward models on AlpacaEval. Meanwhile, experiments on academic Benchmarks show no noticeable regression overall from the starting model.

**Strengths:**

* Novel method of self-generating prompt using persona library as seed.

* Extension of RLAIF work that uses itself as the judge model to perform self-improvement. This is highly efficient and resource-friendly, without needing to collect external feedback or using external reward models.

* It designed a comprehensive alignment experiment, and ablated on size of alignment dataset, and importance of having diversity in prompt space, and optimization algorithm.

* it shows impressive result, especially for Gemma-2 model.

**Weaknesses:**

* "data-free, annotation-free framework " is in accurate. For example, it uses Persona Library as seed, and it uses self-judge to annotate data. Better claim to be free of external annotator and generate prompt space from seed.

* Lack of novelty in the method. It is overall an extension of existing RLAIF literature, using self-judge plus on-policy sampling dataset. The main difference is that it proposes to self-generate the prompt space using Persona library as seed.

* If one major point of novelty in the paper is Persona based prompt space generation, then, experiment will be needed to compare it against existing open-source prompt space, such as UltraFeedback or HelpSteer. I encourage authors to include such baselines to your main experiment tables.

* If the Self-judge portion is the major novelty of the paper, then, experiment should be performed to compare self-judge vs external trained reward or other LLM-as-a-judge reward. An experiment table demonstrating the LLM-as-a-judge's reward accuracy would be helpful to understand the contribution here.

* Evaluation could be improved to be more consistent and clear by comparing AlpacaEval, ArenaHard, and MT-Bench of baseline models + your method all in Table-1, similar to the presentation in SimPO paper. I would like to see how much improvement proposed method brings on all three Benchmarks compared to other methods. Placing heavy emphasis on AlpacaEval2.0 risk over-fitting to one Benchmark.

**Questions:**

* Why is there  such a big difference between SAO's effectiveness on Llama-3 and on Gemma-2? Is it possible that difference arises from that Gemma-2 is a much better reward model when being prompted to pick responses?

An experiment comparing the LLM-as-a-judge accuracy between Llama-3 and Gemma-2 model may be helpful to answer the question.

---

> ### Author Response · Authors · 2024-11-14
>
> Thank you for your valuable feedback. We would like to address your concerns as follows:
>
>
> > **Regarding the novelty of the method and its relation to RLAIF:**
>
> We respectfully disagree with the opinion that SAO is simply an extension of the existing RLAIF literature. The primary novelty of our method lies in the introduction of a new self-improvement framework that aligns LLMs with human performance, without the need for external datasets, human labor, or larger reward models that are generally used in RLAIF framework. This makes our approach highly cost-efficient and resource-friendly. As you noted in your "Strength" comment, this approach eliminates the need to collect external feedback or rely on external tools, which is a significant step forward in self-improvement for LLMs.
>
> ---
>
> > **Regarding the Persona-based prompt space generation and comparison to existing prompts (e.g., UltraFeedback):**
>
> Thank you for suggesting this comparison. To explore the influence of persona role-playing, we included an ablation study in Section 5.4.3 (Figure 3e). Without persona-based generation, even with more repetitive prompts, the model still maintains performance. Additionally, we performed further experiments comparing different prompt sources using the Gemma2-9B-it model with a 10k dataset, shown in the table below:
>
> | **Prompt Source**          | **Win Rate** |
> |----------------------------|--------------|
> | Persona generation         | 72.30        |
> | Random generation          | 62.5         |
> | UltraFeedback              | 55.84        |
>
> This experiment demonstrates that directly using existing prompts (such as those from UltraFeedback, which include objective questions like math and code) can hinder alignment performance. We believe this happens because objective questions might lead the model in the wrong direction, whereas persona- and random-generated prompts (which are subjective) better align the model with human preferences.
>
> ---
>
> > **Regarding the analysis of self-judge portion:**
>
> We appreciate your suggestion to compare self-judging against external reward models. However, in Section 5.4.4 (Figure 3f), we have compared self-judging with random judgment and an external reward model (ArmoRM). Our results show that self-judging is more effective at identifying the correct direction for preference optimization.
>
> ---
>
>
> > **Regarding the performance difference between SAO on Llama-3 and Gemma-2:**
>
> The observed difference in performance between Llama-3 and Gemma-2 is due to the fact that the prompt, response, and judgment processes differ for each model within the SAO framework. A more "intelligent" model, like Gemma-2, is better equipped to conduct self-improvement, which may explain the better results seen with this model. Additionally, we believe that larger models (≥70B) have further potential in this self-improvement process. As for LLM-as-a-judge accuracy, since we focus on subjective prompts (e.g., preferences and opinions) rather than objective tasks like math or code, it’s difficult to define "accuracy" in the traditional sense. The effectiveness of the judge is more about guiding the preference optimization process rather than strict correctness.
>
> ---
>
> We hope these clarifications address your concerns or misunderstandings. If you have any further questions or concerns, we would be happy to discuss them in the rebuttal stage.

---

> ### Comment · Reviewer_acKt · 2024-11-25
>
> Thank you for the response. The comparison against UltraFeedback prompts makes the person-based generation much more persuasive and convincing
>
> However, the experiment setup for self-judge is still unclear, and the claim that the vanilla prompting result is much better than a well-trained reward classifier "ArmoRM" is quite difficult to believe without extensive study of the reward classifying performance. Like what are the areas the self-judge worked better than ArmoRM whcih resulted in the improved performance in the current setup.
>
> The result of paper is promising, but the idea and scientific contribution of the paper needs better scoping. So, I tend to hold my current rating.

---

### Official Review · Reviewer_85Qg · 2024-11-04

**Soundness:** 2
**Presentation:** 2
**Contribution:** 1
**Rating:** 3
**Confidence:** 3

**Summary:**

This paper proposes a way of self-improving without any data annotation, specifically, they randomly sample some personas from Persona-Hub (Chan et al. 2024), and ask base LLMs to produce prompts, then, generate two responses for each prompt, followed by prompting LLMs again for pair-wise ranking. With those data, they can run SimPO to update base LLMs. Experiments show significant improvements on AlpacaEval 2.0 and Arena-Hard Benchmarks, however, there is no difference in many downstream NLP tasks (e.g. GSM8K, MMLU etc.)

**Strengths:**

1.	The improvements on AlpacaEval 2.0 and Arena-Hard Benchmarks over base models are surprising.

**Weaknesses:**

1.	Many researchers are working on self-improving with external supervision and guidance to achieve some improvements over base policy models. And I believe all those studies have already confirmed that LLMs itself cannot be self-improved without using external supervision.
2.	The idea of this paper is very simple, the improvements on AlpacaEval 2.0 and Arena-Hard are very surprising, further investigation and analysis must be done on those results.
3.	The novelty of this paper is very limited.

**Questions:**

The experimental results must be well-examined: a) why the improvements on AlpacaEval 2.0 are so high, but there is no improvement in average score on downstream NLP tasks; b) Why experiments in Table 2 are in few-shot settings? Those models are all instruct models, please test in zero-shot setting; c) Larger LLMs, at least 70B, are required to support the claims of this paper.

---

> ### Author Response · Authors · 2024-11-14
>
> ---
>
> Thank you for your thoughtful comments. We would like to address some of your concerns and misunderstandings as follows:
>
> > **Regarding the ability of LLMs to self-improve without external supervision:**
>
> We respectfully disagree with the assertion that LLMs cannot self-improve without external supervision. In fact, recent work, such as (1) boosting reward models with preference-conditional multi-aspect synthetic data generation and (2) Self-Rewarding Language Models, demonstrates that LLMs can indeed leverage self-generated signals for self-improvement. Our approach builds on this foundation, showing that self-improvement can be achieved through self-supervised signals rather than relying solely on external supervision.
>
> ---
>
> > **Regarding the simplicity of the idea and the surprising improvements on AlpacaEval 2.0 and Arena-Hard:**
>
> While the core concept may appear simple, we believe the effectiveness of the SAO framework lies in its novel combination of dataset-free and annotation-free alignment techniques, which have not been fully explored before. The improvements seen on AlpacaEval 2.0 and Arena-Hard are significant, and we have provided a detailed analysis in Section 5.4 to explain the factors contributing to these results. This includes the influence of dataset size, different preference optimization algorithms, persona effectiveness, and self-judgment ability.
>
> ---
> > **Regarding the experimental results and improvements on AlpacaEval 2.0:**
>
> We have included a detailed explanation in lines 427-431 to clarify this point. Our primary goal was not to generate task-specific data pairs (such as for math or code), but rather to maintain downstream NLP performance while improving alignment ability. While models like Gemma-2-9B-it-SimPO, which are trained on specific datasets, sacrifice some downstream performance to achieve better alignment, our method offers a more balanced approach. SAO demonstrates proactive value by improving alignment without sacrificing downstream task performance.
>
> ---
>
> > **Regarding the use of few-shot settings in Table 2:**
>
> We follow the official few-shot evaluation settings used in the OpenLLM leaderboard, which are the standard in the field and have been adopted by works like SimPO (NeurIPS 2024). We understand that you are suggesting zero-shot settings; however, we believe that comparing with the few-shot setting is more representative of current practices, and we wanted to align with the prevailing methodologies used in the evaluation of models.
>
> ---
>
> > **Regarding the need for larger LLMs (at least 70B) to support the claims of this paper:**
>
> While we acknowledge that larger models can offer potential benefits, it is important to note that in academic research, especially in alignment, working with models of less than 10B parameters is more common due to the constraints of training large-scale models. Despite this, the performance we demonstrate with smaller models has been shown to outperform some of the most advanced models, such as GPT-4. This highlights the practical value of our approach, which can achieve competitive alignment performance without the need for enormous model sizes.
>
> ---
>
> We hope that these clarifications address your misunderstandings and concerns and provide a clearer understanding of our contributions.
>
> ---

---

> > ### Comment · Reviewer_85Qg · 2024-12-02
> > **Official Comment by Reviewer 85Qg**
> >
> > Thanks for providing the detailed explanation.
> >
> > I will maintain my assessment and adjust the soundness rating to better reflect the clarifications provided.

---

### Official Review · Reviewer_ckgo · 2024-11-04

**Soundness:** 2
**Presentation:** 3
**Contribution:** 2
**Rating:** 6
**Confidence:** 2

**Summary:**

This paper presents Self-Alignment Optimization (SAO), a framework that bypasses the need for expensive human-annotated datasets or AI-labeled preference data in aligning LLMs. Instead of relying on annotated data or costly labeling with larger models, SAO aims to replace with a self-supervised alignment process. In SAO, a chat-based model generates its own prompts and feedback through persona-based role-play, evaluates its responses, and refines itself based on these self-assigned preferences. Experiments show that models trained with SAO outperform baseline models on various benchmarks (e.g., AlpacaEval 2.0, MT-Bench, Arena-Hard) in alignment tasks.

**Strengths:**

- The proposed Self-Alignment Optimization (SAO) framework removes the need for expensive human annotations or AI-labeled preference data. By doing so, SAO makes model alignment significantly more affordable and accessible, lowering the barrier for effective model refinement.
- SAO leverages persona role-play to generate a wide variety of prompts, which enriches the diversity of inputs and responses the model can handle. This method shows effective results in generating robust, varied prompts, as demonstrated by the paper's experiments.
- SAO’s effectiveness is shown across different language models and multiple benchmarks (e.g., AlpacaEval 2.0, MT-Bench, Arena-Hard). The framework consistently improves performance across diverse tasks and evaluation settings, indicating its general applicability and robust alignment capabilities.

**Weaknesses:**

- **Dependence on Initial Model Performance for Self-Alignment:** The SAO framework appears to rely on a certain level of initial model performance for effective self-alignment, as the model must be able to understand and follow instructions to generate meaningful responses. However, the paper only demonstrates SAO’s efficacy with relatively well-performing LLMs. It remains unclear how SAO would perform with smaller LLMs, such as those with 1B or 3B parameters, which might struggle with instruction following and generating coherent responses. Further investigation is needed to evaluate SAO’s effectiveness on these smaller-scale models.
- **Challenges with Output Consistency in Self-Supervision:** In some cases, models may fail to produce outputs in the desired format, which could lead to instability or even failure in the training process. This issue poses a risk to the reliability of the self-alignment process. How do authors address instances where the model’s output deviates from the required format during training—whether through corrective measures, filtering techniques, or additional constraints to ensure format consistency.
- To assess the effectiveness of persona-style prompt generation, how does the model's performance compare when using existing prompts instead of persona-based prompts, while keeping all other components of the method unchanged?

**Questions:**

Is there a specific reason for using different prompts for MT-Bench and AlpacaEval?

---

> ### Author Response · Authors · 2024-11-14
>
> Thank you for your valuable feedback. We would like to address your concerns as follows:
>
>
> > **Regarding the dependence on initial model performance for self-alignment (small LLMs):**
>
> We understand your concern about the generalization of the SAO framework to smaller models (1B to 3B parameters). To address this, we conducted additional experiments with the Llama-3.2-Instruct-3B model, and the results are shown below:
>
> | **Model**                    | **Length-Controlled Win Rate** | **Win Rate** |
> |------------------------------|-------------------------------|--------------|
> | Llama-3.2-Instruct-3B        | 21.87                         | 24.97        |
> | Llama-3.2-Instruct-3B-SAO    | 27.44                         | 38.01        |
>
> These results show consistent improvement with the SAO framework, even with smaller models. However, we acknowledge that larger and more capable base models tend to yield better performance, which leaves further potential for larger models (≥70B). This highlights the scalability of the SAO framework as we move to more powerful models.
>
> ---
>
> > **Regarding challenges with output consistency in self-supervision:**
>
> Thank you for raising this important concern. We understand that output consistency is crucial for the success of the self-alignment process. To address instances where the model's output deviates from the desired format, we use regular expressions to extract the relevant output structure, as demonstrated in the code provided in our supplementary material. From our observations, almost all data pairs can be successfully extracted, and we encourage reviewers to run the provided code to verify this process.
>
> ---
>
> > **Regarding the effectiveness of persona-based prompt generation compared with existing prompts:**
>
> Thank you for suggesting this comparison. To assess the influence of persona-style prompt generation, we included an ablation study in Section 5.4.3 (Figure 3e). The study shows that, even without persona-based generation, the model maintains a reasonable level of performance. Additionally, we conducted further experiments comparing different prompt sources using the Gemma2-9B-it model with a 10k dataset, as shown in the table below:
>
> | **Prompt Source**         | **Win Rate** |
> |---------------------------|--------------|
> | Persona generation        | 72.30       |
> | Random generation         | 62.5         |
> | UltraFeedback (existing prompts)            | 55.84        |
>
> These results suggest that using persona-based prompts yields better alignment performance than using existing prompts like those from UltraFeedback, which include objective questions like math and code that may lead the model in the wrong direction using self-generation for optimization.
>
> ---
>
> > **Regarding the use of different prompts for MT-Bench and AlpacaEval:**
>
> We follow the official evaluation scripts and prompts from the FastChat and AlpacaEval GitHub repositories, which are the most current works in the field like SimPO(NeurIPS24). This ensures consistency with standard evaluation practices and enables fair comparison to previous methods.
>
> ---
>
> We hope these clarifications address your concerns. If you have any further questions or suggestions, we would be happy to discuss them in the rebuttal stage!

---

### Official Review · Reviewer_ZirU · 2024-11-04

**Soundness:** 2
**Presentation:** 2
**Contribution:** 1
**Rating:** 3
**Confidence:** 4

**Summary:**

* The paper proposes Self-Alignment Optimization (SAO), a framework for fine-tuning language models without external labeled datasets by leveraging self-generated prompts and responses through persona role-play.
* The study evaluates SAO on two main models: Gemma-2-9B-it and Llama-3-Instruct-8B, testing them on multiple benchmarks including AlpacaEval 2.0, MT-Bench, and Arena-Hard.
* Key findings suggest performance improvements on subjective benchmarks (e.g., Gemma-2-9B-it-SAO achieves 69.2% LC and 66.0% WR on AlpacaEval 2.0) while maintaining performance on objective downstream tasks.

**Strengths:**

* The approach demonstrates meaningful performance gains without requiring external labeled datasets, which could make fine-tuning more accessible and cost-effective for smaller organizations.
* The paper provides nice ablation studies and analyses, particularly in Figure 3, exploring various aspects like dataset size impact, optimization algorithms, persona role-play influence, and self-judgment capabilities.
* The experimental evaluation is systematic, covering both subjective and objective benchmarks, with clear performance metrics and comparisons against baseline models.

**Weaknesses:**

* The paper's technical novelty is limited, as similar self-improvement approaches have been explored in previous works like Self-Rewarding LMs. The core concept appears to be a variation of existing methods rather than a new approach.
  * The claimed benefits of being "dataset-free and annotation-free" are not novel in the field
  * The paper lacks comparison and discussion with relevant prior work on self-alignment, for example SALMON, OAIF, and SAMI.
* There are several methodological concerns:
  * The use of GPT-4o-mini as a judge model for evaluating Arena-Hard and MT-Bench seems questionable given its relative weakness compared to state-of-the-art models
  * The reported improvements on the Open LLM Leaderboard are minimal to non-existent

**Questions:**

* Given that dataset-free or annotation-free alignment approaches are already established in the field, could the authors clearly articulate in one sentence what they consider to be the primary novel contribution of this work?

---

> ### Author Response · Authors · 2024-11-13
>
> ---
>
> Thank you for your feedback. We would like to clarify some points as follows:
>
>
> > **Regarding “the variation of existing approaches” of the paper and several related works**
>
> We appreciate your comments regarding the novelty of our work. However, we respectfully disagree with the suggestion that our proposed method, SAO, is merely a variation of existing approaches such as Self-Rewarding LMs. The overall workflow of SAO is significantly different, particularly with components like persona-based diverse prompt generation and pairwise self-judgment. Specifically, Self-Rewarding LMs rely on external datasets as seed datasets and require a few-shot setting to generate prompts. In contrast, SAO is fully dataset-free and annotation-free, which means it does not require external data for initialization or labeling. As shown in the Self-Rewarding LMs paper, their method performs poorly with only synthetic data, and combining it with external datasets yields better results. This highlights the distinction between our work and theirs.
>
> We also thank you for pointing out relevant prior works such as SALMON, OAIF, and SAMI. We will include a discussion of these works in the Related Work section to better highlight how SAO differentiates itself. However, to better demonstrate the effectiveness of SAO, we focused on comparing it with modern baselines (e.g., Gemma-2-9B-it-SimPO and Gemma-2-9B-it-SPPO) in our main tables.
>
> ---
>
> > **Regarding the use of GPT-4o-mini as a judge model for Arena-Hard and MT-Bench:**
>
> We appreciate your concern about using GPT-4o-mini as a judge model for Arena-Hard and MT-Bench. The primary goal of using GPT-4o-mini was to evaluate the models in a cost-effective manner. GPT-4o-mini has demonstrated reasonable judgment capability in many existing works. To address your concern, we also evaluated Arena-Hard using GPT-4-Turbo as a judge, and the results show consistent improvement: Gemma-2-9B-it: 40.8 vs. Gemma-2-9B-it-SAO: 54.3. We will include these results in the appendix for further clarification.
>
> ---
>
> > **Regarding the reported improvements on the Open LLM Leaderboard:**
>
> We understand your concern that the reported improvements on the Open LLM Leaderboard appear minimal. However, we want to clarify that the primary objective of SAO is not to significantly improve downstream NLP performance but to enhance alignment performance without decreasing the performance of the model.
>
> When comparing SAO to Gemma-9B-it-SimPO, SAO demonstrates comparable alignment performance while maintaining or slightly improving downstream task performance, as shown in Table 2. In contrast, Gemma-9B-it-SimPO, which relies on an external labeled dataset, sacrifices downstream performance for alignment improvements. This trade-off is a critical consideration for real-world applications, and SAO addresses this issue by maintaining performance across tasks.
>
> ---
>
> > **Regarding the primary novel contribution of the work:**
>
> To clarify, the primary novel contribution of SAO is that it introduces a dataset-free and annotation-free framework for self-improvement in alignment, which incorporates persona-based prompt generation and pairwise ranking annotation-free approaches—both of which, to the best of our knowledge, have not been proposed previously in the field. The SAO framework can be applied to various preference optimization methods, such as SimPO, and achieves promising performance comparable to models trained on external datasets, while maintaining downstream task performance. This framework enhances the model’s intrinsic capabilities and ensures consistent performance across a wide range of tasks without significant trade-offs.
>
> ---
>
> We hope these clarifications address your concerns and provide a clearer understanding of our work.

---

### Official Review · Reviewer_TNCW · 2024-11-04

**Soundness:** 2
**Presentation:** 2
**Contribution:** 1
**Rating:** 3
**Confidence:** 2

**Summary:**

This paper proposes Self-Alignment Optimization (SAO), a framework for enhancing language model alignment without external datasets or human-labeled feedback. Traditional alignment methods, like Reinforcement Learning from Human Feedback (RLHF) and Reinforcement Learning from AI Feedback (RLAIF), are resource-intensive due to the need for specialized reward models or substantial external data. SAO addresses these limitations by enabling models to align autonomously: a model generates prompts based on various personas, produces responses, and then ranks these responses through self-assessment. The framework relies on this self-generated dataset and uses preference optimization (SimPO) to refine model alignment, reducing the need for human input or costly external tools.

Empirical results show that SAO achieves improvements in model performance across several benchmarks. For instance, on AlpacaEval 2.0, SAO-tuned models exhibit significant gains in Length-Controlled Win Rate (LC) and Win Rate (WR) over baseline models, with similar enhancements observed on the MT-Bench and Arena-Hard benchmarks. Additionally, SAO-tuned models maintain or slightly improve downstream NLP performance metrics as measured by the Open LLM Leaderboard, suggesting that SAO can enhance alignment without compromising general task capabilities. Further analysis in the paper explores factors that influence SAO's effectiveness, including synthetic dataset size, persona role diversity, and optimization methods, underscoring the potential of SAO as an efficient and scalable alternative to traditional alignment approaches.

**Strengths:**

1. The SAO framework operates without the need for external, human-annotated data, eliminating costly annotation requirements while maintaining alignment effectiveness.
2. Empirical results show that models fine-tuned with SAO achieve substantial improvements over their vanilla base models, particularly in alignment-specific metrics like Length-Controlled Win Rate (LC) and Win Rate (WR) across multiple benchmarks.
3. The paper provides comprehensive empirical results, evaluating SAO across a range of benchmarks and scenarios, including AlpacaEval 2.0, MT-Bench, and Arena-Hard, which thoroughly demonstrate the robustness and scalability of the proposed method.

**Weaknesses:**

1. Most of SAO’s components—such as persona-based prompt generation, self-generation/self-judgment, and preference optimization using SimPO—are adapted from prior work, potentially limiting the novelty of the technical approach. To strengthen the contributions, the paper needs to explore innovative adaptations or improvements to these existing methods that demonstrate clear enhancements in model performance or alignment capabilities.
2. While SAO achieves improvements over vanilla models, its enhancements compared to more meaningful direct counterparts (e.g., Gemma-2-9B-it-SAO vs. Gemma-2-9B-it-SimPO) are relatively minor. Given that SimPO relies on external datasets, this limited improvement suggests that self-generated preference data may currently be less effective. The study needs a deeper examination of self-generated data’s limitations and a discussion of methods to enhance its quality, which may reveal additional advantages over using external data.
3. Although the paper promotes SAO’s dataset-free and annotation-free aspects, the practical impact of this approach is not fully substantiated. Alignment research aims for impactful, real-world outcomes, and it would be helpful if the authors clarified specific advantages or applications where being dataset-free provides measurable benefits (e.g., scenarios with limited access to annotation resources or environments needing continual self-adaptation). For example, people probably don't need a full dataset/annotation-free pipeline, but just a single significant improvement in the RLHF pipeline, like the SimPO technique. Further analysis of the unique applications and limitations of a dataset-free strategy could better ground this approach in practical relevance.

**Questions:**

1. Suggestion on the writing: reduce the length of section 3, which is commonsense knowledge in the field now
2. Questions about the efficiency: the proposed techniques still require the significant cost of training/fine-tuning, how this paper should be positioned in the current trend of much cheaper alignment techniques, like using prompting, representation editing, or reward-guided test-time decoding to improve the alignment performance

---

> ### Author Response · Authors · 2024-11-13
>
> ---
>
> Thank you for your constructive feedback. We would like to clarify some points as follows:
>
>
> > **Regarding the novelty of SAO's components:**
>
> We appreciate your feedback. The motivation of our paper is to propose a novel self-alignment framework that is dataset-free and annotation-free. While we respectfully disagree with the assessment of novelty in terms of individual components, we also provide a detailed analysis of each in Section 5.4. Specifically:
> - **Persona-based prompt generation**: As shown in Section 5.4.3, even without persona role-playing (i.e., using more repetitive prompts), the model still shows significant improvement based on the SAO framework. However, incorporating persona role-playing allows for more diverse prompts, which further enhances alignment performance.
> - **Self-judgment**: While we adopt criteria from Shen et al. (2024) (shown in Fig.1), we introduce a novel pairwise self-judgment method in Section 5.4.4, where our results show that our model outperforms baselines like Random-Judge and ArmoRM-Judge.
> - **Preference optimization**: As demonstrated in Section 5.4.2, SAO is flexible in improving vanilla models using various optimization algorithms, including DPO, ORPO, and SimPO, with SimPO providing the best results, which we adopt as the default in our framework.
>
> ---
>
> > **Regarding the performance improvement compared to direct counterparts trained on external datasets (e.g., Gemma-9B-it-SimPO):**
>
> We understand your concern regarding the quality difference between self-generated datasets and externally labeled datasets. While human-level labels and curated datasets do have better alignment potential, as is commonly acknowledged, our paper introduces a self-improvement framework that operates without external signals. Therefore, it is more intuitive and reasonable to compare SAO against vanilla models (e.g., Gemma-2-9B-it), as the model improves itself in this case. When comparing our SAO method with Gemma-9B-it-SimPO, SAO demonstrates **comparable alignment performance**, while **maintaining or slightly improving downstream task performance**, as shown in Table 2. In contrast, Gemma-9B-it-SimPO, which relies on an external labeled dataset, sacrifices downstream performance for alignment improvements—this is a crucial consideration for practical applications. Additionally, as SAO is still in an emerging research area, we have included the limitations of our framework in Section 7, where we acknowledge that although our approach has shown promising results with simple prompt templates, exploring more complex templates may lead to further improvements.
>
> ---
>
> > **Regarding the practical impact of being dataset-free:**
>
> We acknowledge the importance of improving RLHF pipelines; however, the primary focus of SAO is **data-center efficiency** during the pre-RLHF stage, where it eliminates the need for expensive human labeling or GPT-4-level labor and filtering processes. We believe this approach makes a significant contribution to the field of self-improvement through synthetic data. It offers a new perspective on how self-generated data can enhance LLM alignment with minimal external input. The importance of self-synthetic data is also highlighted in the works we compare to, such as Self-Rewarding and SPPO.
>
> ---
>
> > **Regarding efficiency and cost concerns:**
>
> Thank you for raising this important point. We would like to clarify that SAO does not introduce additional training costs compared to existing RLHF methods (e.g., SimPO). Our focus is primarily on **training dataset construction** rather than increasing training costs. The techniques you mentioned like prompting, representation editing, and reward-guided test-time decoding are aimed at improving the decoding stage and are not directly comparable to SAO. These methods generally focus on training-free approaches, which still lag behind in performance compared to SAO and other mainstream RLHF methods. Furthermore, these decoding-optimation models are not as easily deployable as Hugging Face models.
>
> ---
>
> We hope these clarifications address your concerns and provide a clearer understanding of our work! Please feel free to reach out with any further questions during the rebuttal stage.

---

> > ### Comment · Reviewer_TNCW · 2024-11-28
> > **Thanks for the rebuttal**
> >
> > I thank the authors for the rebuttal. As the authors have clearly identified that their "focus is primarily on training dataset construction," this falls into the synthetic data generation for self-alignment, and this paper has a serious lack of comparison with the abundant self-alignment works, such as [1-3]. I hope the authors can further improve the technique novelty and enrich the baseline comparisons.
> >
> > [1] Sun, Z., Shen, Y., Zhou, Q., Zhang, H., Chen, Z., Cox, D., ... & Gan, C. (2024). Principle-driven self-alignment of language models from scratch with minimal human supervision. Advances in Neural Information Processing Systems, 36.
> >
> > [2] Sun, Z., Shen, Y., Zhang, H., Zhou, Q., Chen, Z., Cox, D. D., ... & Gan, C. (2024). SALMON: Self-Alignment with Instructable Reward Models. In The Twelfth International Conference on Learning Representations.
> >
> > [3] Singla, S., Wang, Z., Liu, T., Ashfaq, A., Hu, Z., & Xing, E. P. (2024). Dynamic Rewarding with Prompt Optimization Enables Tuning-free Self-Alignment of Language Models. arXiv preprint arXiv:2411.08733.

---

### Meta-Review · Area_Chair_xA8e · 2024-12-18

**Metareview:**

This paper proposes a human-feedback-free approach to aligning language models based on data generated from a prompted model. The generated responses are ranked by another prompted model, and the winning/losing responses are used as part of a SimPO objective to align the policy. While this paper tackles an important problem, this space has been explored in prior works, and there is a lack of novelty compared to such works. There is moreover no empirical comparison against such works.

**Additional Comments On Reviewer Discussion:**

Many reviewers noted the lack of novelty against existing works, which I agree with. The empirical improvements over baselines are also marginal. While I appreciated the author rebuttals but they did not change the reviewers' (nor my) assessment of the paper.

---

### Decision · Program_Chairs · 2025-01-22

Reject